# Multipotential Role of Growth Factor Mimetic Peptides for Osteochondral Tissue Engineering

**DOI:** 10.3390/ijms23137388

**Published:** 2022-07-02

**Authors:** Maria Giovanna Rizzo, Nicoletta Palermo, Ugo D’Amora, Salvatore Oddo, Salvatore Pietro Paolo Guglielmino, Sabrina Conoci, Marta Anna Szychlinska, Giovanna Calabrese

**Affiliations:** 1Department of Chemical, Biological, Pharmaceutical and Environmental Sciences, University of Messina, Viale Ferdinando Stagno d’Alcontres, 31, 98168 Messina, Italy; mgrizzo@unime.it (M.G.R.); nicolettapalermo.biologo@gmail.com (N.P.); salvatore.oddo@unime.it (S.O.); salvatore.guglielmino@unime.it (S.P.P.G.); 2Institute of Polymers, Composites and Biomaterials—National Research Council, Viale J. F. Kennedy 54, Mostra d’Oltremare, Pad. 20, 80125 Naples, Italy; ugo.damora@cnr.it; 3Department of Chemistry “Giacomo Ciamician”, University of Bologna, Via Selmi 2, 40126 Bologna, Italy; 4Department of Biomedicine, Neuroscience and Advanced Diagnostics (Bi.N.D.), University of Palermo, Via del Vespro, 129, 90127 Palermo, Italy; martaanna.szychlinska@unipa.it

**Keywords:** osteoarthritis, cartilage, tissue regeneration, tissue engineering, biomimetic peptides, phage-based functional peptides

## Abstract

Articular cartilage is characterized by a poor self-healing capacity due to its aneural and avascular nature. Once injured, it undergoes a series of catabolic processes which lead to its progressive degeneration and the onset of a severe chronic disease called osteoarthritis (OA). In OA, important alterations of the morpho-functional organization occur in the cartilage extracellular matrix, involving all the nearby tissues, including the subchondral bone. Osteochondral engineering, based on a perfect combination of cells, biomaterials and biomolecules, is becoming increasingly successful for the regeneration of injured cartilage and underlying subchondral bone tissue. To this end, recently, several peptides have been explored as active molecules and enrichment motifs for the functionalization of biomaterials due to their ability to be easily chemically synthesized, as well as their tunable physico-chemical features, low immunogenicity issues and functional group modeling properties. In addition, they have shown a good aptitude to penetrate into the tissue due to their small size and stability at room temperature. In particular, growth-factor-derived peptides can play multiple functions in bone and cartilage repair, exhibiting chondrogenic/osteogenic differentiation properties. Among the most studied peptides, great attention has been paid to transforming growth factor-β and bone morphogenetic protein mimetic peptides, cell-penetrating peptides, cell-binding peptides, self-assembling peptides and extracellular matrix-derived peptides. Moreover, recently, phage display technology is emerging as a powerful selection technique for obtaining functional peptides on a large scale and at a low cost. In particular, these peptides have demonstrated advantages such as high biocompatibility; the ability to be immobilized directly on chondro- and osteoinductive nanomaterials; and improving the cell attachment, differentiation, development and regeneration of osteochondral tissue. In this context, the aim of the present review was to go through the recent literature underlining the importance of studying novel functional motifs related to growth factor mimetic peptides that could be a useful tool in osteochondral repair strategies. Moreover, the review summarizes the current knowledge of the use of phage display peptides in osteochondral tissue regeneration.

## 1. Introduction

Articular cartilage (AC) is an avascular, aneural, alymphatic connective tissue, with a limited ability to regenerate and self-repair. It is principally found in the synovial joints, spine, ribs, external ears, nose and airways [1]. AC is composed of a dense extracellular matrix (ECM), containing water, collagen and proteoglycans, with other non-collagenous proteins and glycoproteins present in lesser amounts and a sparse distribution of highly specialized cells called chondrocytes [2,3]. Its main function is to protect the articular surface of bones from abrasion and facilitate the transmission of loads, providing a smooth lubricated surface for joint movement. Defects in AC and associated catabolic processes are irreparable and lead to debilitating degenerative joint diseases, including osteoarthritis (OA) [4]. OA, the most common form of arthritis worldwide, is characterized primarily by the deterioration of AC, and the nearby tissues such as subchondral bone, in joints, and is the main cause of physical disability among older adults [5]. In this context, the preservation of joint tissues is fundamental to joint health and the improvement of quality of life. At present, clinical treatments can only relieve symptoms, including pain and disability, but cannot stop the progression of degenerative events, nor can they regenerate lost cartilage [6]. Several approaches have been pursued to develop engineered bioscaffolds to stimulate osteochondral regeneration with improved functional properties [7,8,9]. However, the approaches developed so far, based on autologous chondrocyte implantation [10,11], osteochondral transplants [12] and microfracture induction [13], can be used only for cases of partial osteochondral damage. Severe osteochondral damage can only be cured surgically by means of joint replacements, an invasive approach that involves removing part or all of the damaged joint and replacing it with artificial implants. Unfortunately, none of these techniques offers longevity or complete restoration of the damaged joint tissues. Hence, the pressing need to find more effective and resolutive solutions for osteochondral repair, both in early and advanced stages of the disease, has opened new routes in the field of osteochondral tissue engineering (OCTE) [14]. OCTE is a promising procedure that uses key factors such as biomimetic materials, mesenchymal stem cells (MSCs) and biomolecules such as growth factors (GFs) to generate functional tissue that is able to regenerate deteriorated bone and cartilage [15,16,17]. With this in mind, biomaterials play a crucial role in supplying a more advantageous microenvironment for cell survival, proliferation and differentiation, as well as carrying bioactive molecules for mediating the cellular response. Recent advances in materials science and engineering, in terms of biomaterials, processing techniques and cell biology for OCTE, have been widely analyzed [18,19]. Therefore, biomaterials that mimic the natural ECM, providing cells with a three-dimensional (3D) environmental structure, are critical and can support cell viability, proliferation and secretory activities [20]. These biomimetic supports are characterized by optimized structural (porosity and pore interconnections), mechanical (elasticity and stiffness) and biological (biocompatibility and biodegradability) properties. However, when employed alone, scaffolds cannot repair osteochondral defects. They need to be functionalized by incorporating bioactive agents that are able to enhance the reparative event and drive cellular functions [21,22]. To this end, the biomaterials can be modified through chemical and biological functionalization routes, obtaining so-called “smart biomaterials”, which have higher protective and regenerative properties. In this context, phage display (PD) technology represents a powerful tool for the selection of specific bioactive peptides for a plethora of biotergets from antibodies to whole cells [23,24]. In this biotechnology, a gene codifying a protein of interest is inserted into a phage coat protein gene, allowing the expression of specific peptides in the outside coat that are screened towards specific targets [25].

On the basis of this evidence, in the present review we aimed to discuss recent progress in the development of bioactive matrices for OCTE. In this scenario, an important role has been given to peptides, because the main cell signaling “language” in the ECM is mediated through peptide epitopes. Peptides could offer a beneficial environment that is permissive for the regeneration of damaged osteochondral tissue, due to their capability to promote interactions with cells and enhance specific tissue-conductive/inductive abilities of biomaterials in terms of host cell recruitment, attachment, proliferation and differentiation [26,27]. In addition to eliciting a cellular response, these bioactive peptide-based matrices promote specific biological responses and guide cell migration with programmed directionality [26]. There are numerous approaches aimed at using bioactive molecules to gain control over cellular functions. Recently, biomimetic modification of electrospun fibers has gained interest in tissue engineering and a variety of techniques have been developed for this purpose. As peptides on the surface of the electrospun nanofibers are capable of a controlled release and enhanced biological responses, many of them have been used to functionalize polymer surfaces in OCTE, as discussed below. Moreover, the release profile of peptides from the functionalized biomaterials is also briefly discussed. Furthermore, the recent development of PD nanotechnology has provided a powerful technique for the screening of tissue-specific peptides with higher tissue penetration ability and high specificities to cartilage and bone tissues [28]. The evidence summarized in this review suggests that GF-related peptides and PD systems have great potential and represent promising strategies in the field of OCTE.

## 2. Growth Factor-Mimetic Peptides for Osteochondral Regeneration

Over the past years, numerous peptides have been isolated from existing GFs. They represent a unique class of bioactive agents that can be easily chemically synthesized with low cost and few immunogenicity issues and can be suitably modified to improve their features and functionalize biomaterial scaffolds. They can form strong networks and have tunable sizes, functional groups and activities, leading to biomolecules with a great potential for OCTE [29]. Peptides, as biological signaling and GF-mimetic molecules, have a crucial function in modulating cellular activities and tissue regeneration. Indeed, they may regulate many aspects of cellular function, including survival, proliferation, migration and differentiation [30], contributing to ECM synthesis [31]. Due to the essential role of GFs in controlling cellular functions, a wide range of GFs has been studied and tested for therapeutic applications in OCTE [32,33].

These peptides have shown the ability to mimic different functions of proteins naturally involved in osteochondral tissue repair, such as transforming growth factor-β (TGF-β) and bone morphogenetic protein (BMP) mimetic and affinity peptides, cell-penetrating peptides, cell-binding peptides, self-assembling peptides and ECM-derived peptides [34]. Furthermore, research on PD peptides is also emerging. In the next paragraphs, the main peptides commonly employed in OCTE are analyzed (Figure 1).

### 2.1. Transforming Growth Factor-β Mimetic Peptides

The TGF-β superfamily consists of approximately 35 multifunctional molecules and is among the principal regulators of chondrogenesis [35]. TGF-β signaling is involved in several activities in both physiological and pathological conditions. It is responsible for gene expression regulation through a wide array of signaling pathways. Among these, the TGF-β-SMAD pathway is the best-characterized one. TGF-β is secreted as an inactive complex of the TGF-β dimer and its pro-peptide, latency-associated peptide (LAP) and latent TGF-β binding proteins (LTBP). Therefore, the secreted TGF-β requires activation before it can bind to its receptor [36]. Activated TGF-βs bind to the tetrameric receptor complex, comprising two TGF-β type I (TβRI) and two TGF-β type II (TβRII) receptors. TβRII trans-phosphorylases TβRI and activates TGF-β-specific SMADs [37]. TGF-β–SMAD2/3 signaling promotes proliferation, chemotaxis and early differentiation of chondroprogenitor cells (Figure 2). Meanwhile, it inhibits osteoblast maturation, mineralization and transition into osteocytes. However, it has been shown that TGF-β loses its protective effects in cartilage with aging. Indeed, during chondrogenesis there is a shift from SMAD2/3 to the SMAD1/5/8 route activation, which is associated with cell terminal differentiation and hypertrophy. Data from the literature suggest that TGF-β is crucial for the initial steps of chondrogenesis and cartilage integrity, and its absence and/or abnormality in its pathways, resulting in cartilage homeostasis impairment and tissue degeneration, resembling features of the OA phenotype [38]. These results indicate, finally, that TGF-β represents a powerful tool to prevent or repair cartilage damage. However, due to its unstable structure, it is difficult to procure it in a pure form to be used in osteochondral repair. Therefore, using peptides that mimic the structure and function of TGF-β and that are more stable, cost-effective, durable and can be easily administered to patients represents an encouraging strategy in OCTE.

Cytomodulins (CMs) are oligopeptides developed to function similarly to TGF-β and they represent an example of TGF-β mimetic peptides. It has been demonstrated that these oligopeptides, similarly to TGF-β, are able to improve collagen I expression and promote wound healing *in vitro* [39]. Furthermore, Mittal A. et al. showed that the conjugation of CM-1 with a poly(lactide-co-glycolide) (PLGA) scaffold can increase wound healing *in vivo* [40]. In particular, CM-1 was covalently coupled, through its free –NH_2_ groups, to hydrolyzed PLGA microspheres, using pendant –COOH groups, thus forming amide (CONH_2_) bonds. Carbodiimide was used as the coupling agent to covalently attach CMs. The CM solution used for coupling was prepared in acetate buffer (200 ng/mL) and the reaction was continued for 24 h [40]. On the other hand, in contrast to TGF-β, other evidence has demonstrated that CMs are able to induce the chondrogenic differentiation of stem cells only when they are bound to the surface of microspheres and not in soluble form [41,42]. Shah et al. designed a self-assembling biomaterial based on peptide amphiphile (PA) nanofibers conjugated to the TGF-β1 affinity peptide and evaluated its chondrogenic regenerative properties both *in vitro* and *in vivo*. The authors also evaluated the GF release kinetics to determine if the presence of binding epitopes to TGFβ1 on the used biomaterial was able to slow down its release from the gel. The results demonstrated the slower release of the latter when compared to the control, suggesting that successful binding occurred, which may have led to a prolonged release at the defect site and enhanced tissue regeneration. Their *in vitro* studies highlighted that this self-assembling PA scaffold promoted hMSC viability and chondrogenic differentiation. Furthermore, they demonstrated that the scaffold was able to enhance the regenerative potential of microfracture-treated chondral defects in a rabbit model, without the addition of exogenous GFs in the presence of marrow-derived mesenchymal cells [43]. Furthermore, their study showed that the ability of affinity peptides to recruit TGF-β *in vivo* allowed them to bypass problems related to its dilution. In another study by Chen et al. [44], TGF-β1 affinity peptides were incorporated and preloaded within porous chitosan scaffolds with different mass ratios, modified with carboxylic groups using succinic anhydride and successively grafted with the peptide HSNGLPL via the 1-ethyl-3-[3-dimethylaminopropyl] carbodiimide hydrochloride/N-hydroxysuccinimide (EDC/NHS) method, to increase ectopic cartilage formation *in vivo*. The authors concluded that TGF-β1 released from the chitosan 30% sponges presented a slower release curve without an initial burst release of TGF-β1 at the early time-points, suggesting successful binding between TGF-β1 and TGF affinity peptides and the positive effects of the affinity peptide on the controlled release of GFs.

TGF-β1 affinity binding peptides have been also used to produce bilayered porous scaffolds with gelatin methacryloyl hydrogels as a matrix, manufactured via three-dimensional (3D) printing, where the upper layer was covalently bound with TGF-β1 affinity binding peptides that were able to adsorb TGF-β1 and the lower layer was blended with hydroxyapatite for subchondral regeneration. The bilayered scaffolds showed promising therapeutic efficacy, as proven via chondrogenic and osteogenic induction *in vitro* and osteochondral repair *in vivo* [45]. In agreement with the previous study, Ju et al. developed a new photo-crosslinked gelatin methacryloyl hydrogel (GelMA) modified with a TGF-β1-affinity peptide (HSNGLPL), which was able to self-absorb with TGF-β1, and demonstrated that this scaffold was able to promote cartilage repair [46]. The hydrogel showed good elasticity and resistance to pressure, providing suitable conditions for cell adhesion and the cartilage differentiation of MSCs. HSNGLPL was released as GelMA degraded, so the release rate of HSNGLPL was significantly reduced to achieve a longer-term effect of recruiting TGF-β1 *in vivo*. Furthermore, the functionalized peptide hydrogel promoted the formation of collagen type II and reduced the expression of MMP-13, the most effective collagen-type-II-degrading enzyme in the MMP family, to promote the reconstruction of cartilage [46].

Bilayered osteochondral scaffolds were also developed by Wang et al. through cryogenic 3D printing. In particular, osteogenic peptide/β-tricalcium phosphate/PLGA water-in-oil composite emulsions were printed to obtain the porous subchondral layer, whereas poly(D,L-lactic acid-co-trimethylene carbonate) water-in-oil emulsions were processed into a thermal-responsive cartilage frame on top of the subchondral layer. The osteogenic peptide (P24) had a sequence of KIPKA SSVPT ELSAI STLYL SGGC, with a purity of 98.12%. The cartilage frame was further functionalized with TGF-β1-loaded collagen I hydrogel to form the cartilage compartment. The results showed that the two layers were closely bonded together, showing excellent shear strength and peeling strength. The *in vitro* release behavior of P24 and TGF-β1 from both compartments was analyzed over 30 days. In particular, P24 was released from the subchondral zone in a sustained manner, with an initial burst release (25%, 48 h), followed by a slower but steady release (70% in 30 days). On the contrary, TGF-β1 showed a faster profile with a higher initial release (47%, 48 h), followed by a slower but sustained release (90% in 30 days). Rat BMSCs exhibited high viability and proliferation in both layers. Moreover, gradient rBMSC osteogenic/chondrogenic differentiation was also obtained in the osteochondral scaffolds [47].

Furthermore, the SPPEPS peptide, a common sequence between TGF-β3 and aggrecan molecules, has been used for the same application. Indeed, it has been shown that SPPEPS significantly improves collagen II expression *in vitro* and can also enhance collagen XIα1 expression relevant to cartilage formation [48]. According to previous evidence, gene ontology analysis demonstrated that SPPEPS is able to upregulate chondrogenesis-related genes, including *ENPP1* and *CLIC4* [48]. Pentenoate-functionalized hyaluronic acid (PHA) hydrogels, functionalized with both SPPEPS and arginine–glycine–aspartic acid (RGD), significantly enhanced the expression of collagen type II in rat MSCs by about 300-fold when compared to the control group, suggesting the potential of the peptide for chondroinductivity [48].

### 2.2. Bone Morphogenetic Protein 2

Bone morphogenetic protein 2 (BMP-2) is a member of the TGF-β super-family. It plays a pivotal role in the regulation of chondrocyte proliferation and maturation during endochondral bone development [49]. Under an appropriate regulatory profile, BMPs can induce both osteogenic and chondrogenic differentiation of stem cells *in vitro* [50]. Previous studies have demonstrated that BMP-2 has an essential role in different forms of arthritis [51,52,53].

Similarly to the TGF-β superfamily GFs, BMPs bind to transmembrane serine/threonine kinase receptors on the cell membrane, triggering specific intracellular signaling pathways to activate and affect gene transcription [54]. There are two types of BMP receptors, type I and type II, which are both necessary to initiate further signaling events [55]. Upon ligand binding, the SMAD1/5/8 route is activated and promotes almost every step during osteoblast differentiation and maturation. In particular, P-SMAD1, 5 and 8 assemble into a complex with SMAD4 and translocate into the nucleus to regulate the transcription of target osteogenic and chondrogenic genes, such as osteogenic-related genes—inhibitor of DNA binding 1 (Id1), distal-less homeobox 5 (Dlx 5), runt-related transcription factor 2 (Runx2), and osterix—and chondrogenic-related genes—sex-determining region Y box 9 (Sox9) [56]. The regulation of Sox9 expression by the BMP signaling pathway plays an important role in early chondrogenic differentiation and cartilage formation [57]. In addition, BMP also initiates SMAD-independent signaling pathways, resulting in the activation of p38 MAPK and JNK [55].

Since BMP-2 is particularly implicated in the promotion of chondrogenesis, several peptides have been designed to mimic this effect. Casein kinase 2 (CK2) is a highly conserved and ubiquitously expressed enzyme and regulator of BMP-2. Three peptides, CK2.1, CK2.2, and CK2.3, have been designed to activate the BMP signaling pathway [58]. Among them, CK2.1 is derived from CK2 phosphorylation sites generated by the BMP receptor IA at amino acids 466–469 (SYED) and effectively induces the formation of cartilage [59]. In the C3H10T1/2 cell micromass chondrogenesis model, CK2.1 (100 nM) leads to increased proteoglycan expression and SMAD binding element (SBE) luciferase activity, which can be compared to stimulation with 40 nM BMP-2. In addition, CK2.1 leads to higher expression of collagen type II than BMP-2 [60]. The advantage of CK2.1 over BMP-2 is due to the absence of hypertrophy or mineralization induction, evidenced by the lack of collagen type X and osteocalcin expression by CK2.1. The latter was shown to increase cartilage width and collagen type II and IX expression in femoral articular cartilage, but no increase in collagen type X and mineral trabecular bone mineral density was observed after its injection into the tail vein and the knee cartilage defects of C57BL/6J mice [59,60]. In particular, Akkiraju et al. designed hyaluronic-acid-based hydrogel particles (HGPs) functionalized with cysteine-tagged CK2.1 peptide. The authors conjugated the peptide CK2.1 to HGPs through hydrolytically degradable ester linkages (HGP-CK2.1) via the reaction of modified HGPs (10 mg) in 10 mL PBS containing 10 mg of the CK2.1 peptide. This allowed the slow release of the CK2.1 over a 7 day period, highlighting promising results for cartilage repair in a mouse model, which were comparable to sham-operated mice without the induction of chondrocyte hypertrophy [60]. On the contrary, BMP-2 significantly enhanced cartilage width, collagen type X and trabecular bone mineral density, but did not increase collagen type II and IX expression. These data indicate that CK2.1 drives chondrogenesis and cartilage formation without the induction of chondrocyte hypertrophy, suggesting its use as a promising therapeutic tool for cartilage-degenerative diseases. In the same model, peptides CK2.2 and CK2.3 induced osteocalcin expression and mineral deposition as a consequence of osteoblast differentiation, which resembled BMP2 stimulations [59]. Several other peptides mimicking BMPs have been designed for their osteoinductive properties as well. BMPs have two sequences denoted as the “wrist” epitope, which binds to the BMP receptor type I, and the “knuckle”, which binds to the BMP receptor type II. Contained within the knuckle epitope of BMP-2, a 20-mer sequence (NSVNSKIPKACCVPTELSAI) has osteogenic activity [61]. Another peptide of this family, BMP-9, also possesses abundant osteogenic activity. A peptide derived from the knuckle epitope of the BMP-9, known as pBMP-9 elicits the mRNA expression of osteogenic markers and ECM mineralization to a slight degree. These effects were not as pronounced as those of the growth factors at the equimolar concentration. A higher dose could, however, compensate for its lowered activity in prompting the transcription of certain markers but not of all of them [62].

In a study by Zouani et al., different variants of BMP-2 (RKIPKASSVPTELSAISMLYL), such as a BMP-9-derived peptide (RKVGKASSVPTKLSPISILYK) and a BMP-7-derived peptide (RTVPKPSSAPTQLNAISTLYF), were implanted onto RGD-conjugated polyethylene terephthalate (PET) surfaces and compared. In terms of the osteogenic marker expression and the ECM thickness, the BMP-2 peptide showed the highest activity, followed by the BMP-7- and BMP-9-derived peptides [63]. A BMP-2-derived peptide containing the DWIVA pentamer, corresponding to the BMP receptors I and II binding site sequences, is known as the osteopromotive domain. This peptide has been shown to evoke the proliferation and differentiation of MC3T3-E1 osteoblast cells *in vitro* when conjugated to titanium surfaces [64]. In this case, the authors first performed the activation of a titanium surface for the synthetic peptide by means of 3-aminopropyltriethoxysilane (APTES) grafting. Successively, the APTES-treated implant or disc reacted with the heterobifunctional crosslinker, N-succinimidyl-4-(maleimidomethyl)-cyclohexancarboxylate, for 2 h. The implant was then collected, washed and then reacted with the synthetic peptide containing DWIVA (2 mg/0.5 mL in phosphate buffered saline) [64].

Interestingly, Renner et al. found that the BMP-2-derived peptide also showed significant activity in inducing chondrogenesis at a concentration of 100 μg/mL, proving to promote the production of proteoglycans without extensively upregulating hypertrophy as occurs with BMP-2 [65].

Another important peptide is represented by B2A2, which is designed and chemically synthesized and contains two principal functional domains: a BMP receptor-targeting domain and a heparin-binding domain, which is linked by a hydrophobic spacer domain (KK-[NH(CH2)_5_CO]_3_). B2A2 interacts with BMP receptor isoforms, potentiating the action of BMP-2. B2A2 is clinically used to coat ceramic granules used as a bone substitute material to treat end-stage hindfoot arthritis [65]. Surprisingly, it also enhances chondrogenesis. Indeed, in a micromass chondrogenesis model of C3H10T1/2 fibroblast cells, microarray analysis showed that B2A2 significantly enhanced chondrogenesis-related gene expression. Moreover, B2A2 treatment significantly increased cartilage proteoglycans synthesis and chondrocyte density *in vivo* [66].

Interestingly, several studies have been carried out in the context of bioactive electrospun fiber mimetic peptide loading for bone regeneration. For instance, Boda et al. developed mineralized nanofiber segments coupled with calcium-binding BMP-2 mimicking peptides for periodontal bone regeneration [67]. In a more recent study, the authors investigated whether the dual delivery of Alendronate, a calcium-chelating bone therapeutic, and an E7-BMP-2 mimicking peptide incorporated onto mineralized nanofiber fragments of polylactide-co-glycolide-collagen-gelatin, could help to regenerate alveolar bone *in vivo*, suggesting its satisfactory regenerative potential [68]. Weng et al. fabricated 3D hybrid nanofiber aerogels from 2D electrospun fibers and loaded them with heptaglutamate E7-domain-specific BMP-2 mimicking peptides for cranial bone tissue regeneration, demonstrating the osteoinductive property of the aerogel [69]. More recently, Toprak et al. developed a metal-organic framework (MOF)-embedded electrospun fiber scaffold for the controlled release of BMP-6 to enrich bone regeneration efficacy [70]. In a study by Laboy-López et al., a porous cellulose acetate fiber mat was developed using the electrospinning technique and the mats were chemically modified to bioactivate their surface by coupling adhesive osteogenic peptides such as KRSR, RGD and the growth factor BMP-2 to favor osteoconduction [71]. Sun et al. fabricated a cell-free tissue-engineered periosteum based on an electrospinning poly-L-lactic acid (PLLA) nanofiber, where VEGF and BMP-2 were immobilized to enhance the durability of angiogenesis and osteogenesis during bone regeneration in bone with massive periosteum defect regeneration [72].

### 2.3. Cell-Penetrating Peptides

Cell-penetrating peptides (CPPs) are a class of peptides that are able to cross the cellular membrane, transporting their ‘cargo’ into the cytoplasm [73]. Since the chondrogenic differentiation of stem cells can be finely tuned by means of TGF-βs, different studies based on transfection of the related genes into stem cells using adenoviral and lentiviral vectors have been performed [74]. For example, Guo et al. combined the human immunodeficiency virus TAT protein with the SV40 large T protein nuclear localization signal (NLS) to form the NLS-TAT peptide. This peptide was employed to deliver the hTGF-β3 plasmid into pre-cartilaginous stem cells, promoting chondrogenesis [75]. In particular, CPP was employed as a carrier for the hTGF-β3 plasmid on a self-assembled peptide scaffold [76]. Similarly, Jo et al. demonstrated that CPP-conjugated co-activator-associated arginine methyltransferase 1 (CARM1) protein was successfully delivered into human MSCs, changing their global gene expression profile and upregulating their ability to differentiate [77].

### 2.4. Cell-Binding Peptides

Among all cell-binding peptides, PepGen P-15 is a highly stable peptide, characterized by 15 amino acids that are identical to the cell-binding region of collagen type I [78]. When used to functionalize scaffolds, P-15 can improve cell attachment to bone substitutes, upregulating ECM production and significantly increasing the BMP-2 and BMP-7 transcript levels, as well as alkaline phosphatase (ALP) [79,80]. This upregulated gene expression may suggest that P-15 has the potential to promote osteoblastic activity in human osteoblast cells. Indeed, results from different studies have demonstrated that P-15 can stimulate *in vitro* the proliferation and differentiation rate, as well as the GF production, of osteoblasts [81,82,83]. Furthermore, when combined with platelet-derived GF (PDGF), a significant improvement in both proliferation and calcification was effectively observed [84]. Preclinical results have also confirmed that P-15-containing bone graft substitutes can allow bone healing and regeneration [85]. However, some controversial data still exist on this topic. In particular, some authors reported less favorable results with P-15-containing graft substitutes [86,87]. For example, Vordemvenne et al. [84] highlighted that P-15 alone is not capable of upregulating the proliferation and calcifying potential of human osteoblasts *in vitro*. In addition, application of the P-15-containing graft substitutes was found to accelerate the process of the early bone formation response but, at the same time, not the long-term effects [88,89,90,91].

### 2.5. Self-Assembling Peptides

Self-assembly peptides are composed of either alternating hydrophilic and hydrophobic amino acids or peptide amphiphiles [92,93,94], which are able to self-assemble into nanofibers and form nanofibrous hydrogels. Peptide amphiphiles can support osteoprogenitor cells and guide their differentiation [95,96]. Lee et al. demonstrated that mineralized matrices containing peptide amphiphiles promoted the osteogenic differentiation of hMSCs [97]. Among different self-assembly peptides, RADA16-I (AcN-RADARADARADARADA-CONH2) is a synthetic, commercially available peptide (PuraMatrix). Cell adhesion, proliferation and differentiation of osteoblasts were found to be higher in the RADA16-I-containing scaffold [98]. Furthermore, the addition of BMP-2 into a hydrogel RADA16-I functionalized matrix significantly enhanced bone regeneration in an animal bone defect model [99]. In this context, bone regeneration is expected to be affected by the kinetics of the peptide release from scaffolds. In particular, the binding effect of the mentioned matrix is dependent on the structure and electrostatic charge-induced interactions between the protein and the peptide nanofiber. Therefore, the protein release is dependent on protein size, peptide nanofiber density and electrostatic charge [100].

Peptide gels with binding sites for TGF-β1 have been also prepared by synthesizing peptide amphiphile-TGF-β1 affinity peptide conjugates. In such a way, the authors were able to obtain a controlled release system that facilitated the chondrogenic differentiation of encapsulated MSCs *in vitro* within 4 weeks [101].

### 2.6. RGD Sequence Peptides

Arginylglycylaspartic acid (RGD)-derived peptides, designed as a specific recognition site for integrin receptors, which in turn are reported as key regulators of cell–cell and cell–ECM communication, have also been used in OCTE. They have been shown to improve cell adhesion, which is fundamental for long-term cell survival and cartilage and bone tissue repair [101]. In particular, their mechanism of action involves binding to integrins on cell membranes to mediate cell adhesion, spreading and other cell activities [101]. The RGD motif is the minimal sequence and is normally combined with different amino acid linkers. Currently, RGD sequences that are widely used in the field of cartilage and bone repair include GRGDS, RGDS, YRGDS and c(RGDfk) [102]. RGD is normally used to improve the bio-inertness of synthetic materials. Indeed, several studies have reported enhanced cell attachment, vitality and enhanced differentiation with biomaterials coated with RGD peptides [103]. To date, RGD peptides suitably immobilized on different biomaterials such as natural (i.e., alginate collagen, hyaluronic acid, cellulose) and synthetic polymers (polyethylene glycol (PEG) or titanium oxide nanotubes) have been associated with the increased biological performance of MSCs in tissue regeneration. It is worth noting that RGD-containing scaffolds, used to deliver GFs such as BMP-2 to promote osteochondral regeneration in experimental models, have also been used with favorable results [104,105].

### 2.7. N-Cadherin Mimetic Peptides

N-cadherin (NC) is a key factor mediating cell–cell interactions during mesenchymal condensation and chondrogenesis. The introduction of NC mimetic peptides into chemically modified hyaluronic acid-based hydrogels can promote both the early chondrogenesis of MSCs and late cartilaginous matrix production [106]. Indeed, the incorporation of these developmentally relevant cues was possible in hyaluronic acid hydrogels, where provisions for receptor interactions and the inclusion of peptides that mimic the ECM of NC can be engineered into the synthetic microenvironment. Frith et al. in their *in vitro* and *in vivo* studies investigated and blocked these interactions to understand their role in hMSC chondrogenesis and neocartilage production [106]. NC mimetic peptides contain a conserved three-amino acid sequence, His-Ala-Val (HAV), which provides a homophilic cell adhesion recognition site and mediates cell-to-cell adhesion. For example, Bia et al., showed that MSC interactions with modified hyaluronic acid hydrogels promoted chondrogenesis via cell surface receptors CD44 and CD168 [107]. Furthermore, the conjugation of an NC mimetic peptide containing the “HAVDI” sequence, derived from the first extracellular domain of NC, further enhanced the early expression of chondrogenic markers and promoted long-term cartilage matrix production under both *in vitro* and *in vivo* conditions [108]. This finding highlighted the importance of the bio-functionalization of biomaterial scaffolds, with developmentally relevant cues in enhancing the regenerative outcomes of stem-cell-based treatments. More recently, Guo et al. developed a bilayered, modular hydrogel and implanted it *in vivo*, in a rabbit femoral condyle defect model, to investigate its effect on osteochondral tissue repair [109]. Articular cartilage repair was enabled via the bioconjugation of an NC sequence, whereas the subchondral bone repair was enabled via the conjugation of a glycine-histidine-lysine (GHK) sequence. In particular, the crosslinker poly(glycolic acid)-poly(ethylene glycol)-poly(glycolic acid)-di(but-2-yne-1,4-dithiol) (PdBT) was click-conjugated with either a chondrogenic NC (“GGGHAVDI”) or osteogenic GHK (“GGGGHKSP”)–peptide sequence and then mixed with a suspension of thermoresponsive polymer, poly(N-isopropylacrylamide-co-glycidyl methacrylate) (P(NIPAAm-co-GMA) and MSCs to generate tissue-specific, cell-embedded hydrogel layers for cartilage and bone. The hydrogel layers were fabricated by mixing P(NIPAAm-co-GMA); either MSCs or no cells; and either PdBT (control), GHK/PdBT (osteogenic) or NC/PdBT (chondrogenic) in PBS on ice to obtain homogeneous pre-hydrogel suspensions, which were then poured into cylindrical Teflon molds (37 °C; 5% CO_2_, 1.5 h). The authors showed that the hydrogel system combined with N-cadherin mimetic peptides significantly enhanced the osteochondral regeneration compared to controls, demonstrating the efficacy of this system [109].

### 2.8. Laminin-Derived Peptides

Laminins are ECM proteins that bind to cell membranes through integrin receptors and other cell membrane molecules. The peptides isoleucyl-lysyl-valyl-alanyl-valine (IKVAV) and YIGSR, derived from the A and B1 chains of laminin, respectively, promoted MC3T3-E1 attachment to plastic dishes when coated with these peptides. Increasing evidence showed that laminins can influence cartilage and bone regeneration by regulating cell functionality, such as adhesion, migration and survival [110]. The most significant osteogenic differentiation effects were demonstrated by IKVAV when compared to the YIGSR peptide [111]. A more recently discovered laminin-derived peptide, Ln2-p3, has been also shown to enhance the expression of several osteogenic markers and increase the alkaline phosphatase (ALP) activity of cells when coated on titanium surfaces [112]. It has also been shown that chondrocytes are responsible for the production of various laminins during different developmental stages of cartilage formation [113]. Data from the literature demonstrate that specific laminin isoforms incorporated within biomaterials could enhance the proliferation capacity of adult stem cells when cultured on a coated or soluble laminin environment, suggesting their overall positive effect on tissue regeneration [114].

### 2.9. Parathyroid Hormone 1–34 Peptide (Teriparatide)

Parathyroid hormone (PTH) is an 84 amino-acid, naturally occurring protein that plays a major regulatory role in mammalian mineral ion homeostasis. PTH1–34, the peptide derived from its 34 amino-acid domain, displays similar activity to the full-length protein [115]. Among its several functions, PTH1–34 stimulates osteoblast proliferation, differentiation and prevents apoptosis [116]. Evidence from animal models shows that daily subcutaneous injections of PTH1–34 significantly increased the bone mineral content and density, as well as the total osseous tissue volume, torsional strength and stiffness [117]. Additionally, accelerated callus mineralization, increased bone density at the fracture site and better mechanical properties of the united bone have been reported [118,119,120]. Moreover, PTH1-34 has emerged as a promising peptide in the treatment of osteochondral defects as its systemic administration stimulates both articular cartilage and subchondral bone repair [121].

### 2.10. Osteogenic Growth Peptide

Osteogenic growth peptide (OGP), a naturally occurring 14-mer growth factor peptide, is primarily found in serum and promotes bone anabolism, leading to increased bone formation and an overall increase in bone mass. Pigossi et al. designed bacterial cellulose (BC) membranes that incorporated hydroxyapatite and OGP or OGP(10–14) at a final concentration of 10^−9^ mol L^−1^. The BC/hydroxyapatite membrane resulted in efficient bone regeneration in critical-sized bone defects. Nevertheless, the incorporation of peptides did not increase bone regeneration in this study model [122]. Further studies were performed to improve the effectiveness of the peptide release time at the required concentration for longer periods, to obtain *in vivo* results that were as effective as those in previous *in vitro* studies using composites based on BC [123]. Indeed, OGP exhibited a pivotal role in the regulation of osteoprogenitor cell proliferation, ALP activity, collagen production, differentiation, secretion of osteocalcin and ECM mineralization. When dissociated, OGP is proteolytically cleaved, producing a C-terminal pentapeptide YGFGG, known as OGP10–14. This pentapeptide activates the intracellular Gi-protein-MAP kinase pathway. Both OGP and OGP10–14 enhance the early expression of various markers related to osteogenesis, such as Runx2, ALP, osteopontin, osteoprotegrin and osteocalcin, and they have been used to functionalize biomaterials, demonstrating enhanced osteoinductive potential [123].

### 2.11. Other ECM-Derived Peptides

GFOGER (glycine-phenylalanine-hydroxyproline-glycine-glutamate-arginine) is a collagen-mimetic peptide. It selectively promotes α2β1 integrin binding, which is a crucial event for osteoblastic differentiation. Different studies have shown that implants coated with GFOGER improved peri-implant bone regeneration and osseointegration [123]. For example, Reyes et al. established a biologically active and clinically relevant implant coating strategy that enhances bone repair and orthopedic implant integration. To this end, the authors coated a titanium implant with GFOGER, which was diluted to 20 μg/mL in Dulbecco’s PBS and incubated on the titanium surfaces for 1 h at 22 °C [124].

L-α-Aspartylglycyl-L-α-glutamyl-L-alanine (DGEA) is a recognition motif employed by type I collagen to bind to α2β1 integrins. This collagen peptide sequence promotes cell adhesion, spreading and osteogenic differentiation [125]. Reyes et al. designed a triple-helical collagen-mimetic peptide that specifically targeted the α2β1 integrin receptor and promoted density-dependent cell adhesion, focal adhesion formation and FAK phosphorylation, overcoming the classical issues related to the design of biomaterial surfaces and tissue engineering scaffolds using whole ECM molecules, such as type I collagen, which are often limited by a lack of specificity for particular integrins, thus exhibiting minimal control over cellular responses [125].

In another study, Egusa et al. showed that the SVVYGLR (Ser-Val-Val-Tyr-Gly-Leu-Arg) peptide sequence, found adjacent to the RGD sequence in osteopontin, significantly promoted the adhesion and proliferation of human MSCs and endothelial cells. Furthermore, they showed that the synthetic SVVYGLR peptide also suppressed osteoclastogenesis, contributing to bone repair at the early stage [126]. Indeed, the authors implanted a collagen sponge containing the peptide or PBS (control) in standardized bone defects in rat calvariae and they demonstrated that the number of TRAP-positive osteoclasts in the grafted sites after 3 weeks was significantly lower in the peptide group. By the 5th week, significantly enhanced resorption of the grafted collagen sponge and new bone formation was observed within and surrounding the sponge in the peptide group. The authors specified that the grafted collagen sponge was a commercially available material and it was not specifically designed for the controlled release of the peptide. Therefore, it was most likely that the peptide was quickly released from the sponge after implantation, facilitating early attachment and proliferation of mesenchymal osteoprogenitor cells, as well as suppressing migration and maturation of osteoclasts, suggesting that peptide treatment resulted in accelerated remodeling-based bone regeneration.

In Table 1 are reported the most relevant biomimetic peptides used in osteochondral regeneration applications.

## 3. Phage Display Functional Peptides

PD is an innovative approach, in which functional exogenous peptides are exposed on the capsid surface of a bacteriophage as a fusion product with endogenous proteins of the phage coat. The biological vehicle more widely used in PD combinatorial libraries is the filamentous M13 bacteriophage, which contains three main structural protein regions on the virion, pIII, pVIII and pVI. The genes for these three proteins can be manipulated to display random peptides outside the phage. The pIII is the most commonly used gene and it allows the presentation of approximately five copies of the randomly generated peptide [127,128]. In contrast, the pVIII gene is expressed about 2700 times. Thus, a random peptide fused with pVIII will be expressed 2700 times per phage. Random peptide insertions into the pIII/pVIII proteins allow the development of wide libraries which can be used to select highly specific peptides mimicking the structural and functional features of several biological molecules.

This approach is mostly used to identify bioactive peptides that bind specifically to receptors on the stem cell surface and is used to functionalize synthetic scaffolds (2D and 3D hydrogels) for promoting stem cell growth and differentiation in vitro. Further, mimetic peptides selected via PD have been incorporated into injectable biomaterials (e.g., nanofibers), which can bind to GFs and promote the repair of endogenous tissue *in vivo* (Figure 3). Indeed, recently, several groups used PD technology to select tissue-specific peptides able to stimulate cell adhesion, proliferation and differentiation and improve osteo-chondral regeneration [129,130]. For example, Zymolik and Mummert identified a new hyaluronic acid binding peptide (pep-1) via PD that was able to improve the hyaluronic acid production and deposition on the ECM of the cartilage for repair and remodeling [131]. Similarly, Metz-Estrella et al. identified a regulatory protein that modulates osteoblast differentiation, TGFβ receptor-interacting protein 1 (TRIP-1), from the osteoblasts using a PD library [132]. In another study, a transforming growth factor-β1 (TGF-β1)-binding peptide (HSNGLPL) was identified and integrated into nanofiber gel materials to promote the chondrogenic differentiation of MSCs *in vitro* and the repair of a chondral defect *in vivo* [133].

Cabanas-Danés et al. developed a bifunctional peptide platform for the targeting of collagen type II and the delivery of TGF-β1 to stimulate chondrogenic differentiation *in vitro*, increasing glycosaminoglycan (GAG) production [134]. 

Moreover, the BMP-2-binding peptide (TSPHVPYGGGS) was obtained via the biopanning PD technique and integrated with peptide amphiphile (PA) nanofibers to generate a gel scaffold for bone tissue engineering. The results showed that the bioactive nanofiber system was able to improve osteogenesis both *in vitro* and *in vivo*, suggesting its promising application in bone grafting [43,135,136].

All these results highlight that PD is a powerful tool to identify bioactive molecule-binding peptides to induce the osteochondral regeneration of stem cells.

## 4. Discussion and Conclusions

The application of biomimetic peptides has allowed researchers to achieve significant results related to osteochondral regeneration both *in vitro* and *in vivo*. The use of PD to identify GF-related peptides capable of stimulating cell adhesion, proliferation, differentiation and migration could offer a further alternative approach to the OCTE strategy. These functional peptides have several advantages, such as (i) high selectivity and specificity against the target of interest; (ii) easy production and cost-effective synthesis both on a small and large scale; (iii) high bio-activity and absorbability; (iv) good biodegradability and biocompatibility; (v) their addition to the scaffolds without any chemical reaction by co-visualizing both the functional peptide and the material-binding peptide of the scaffold on the phage surface, promoting both growth and differentiation; (vi) the possibility of loading them with drugs and targeting them at specific cellular receptors for drug delivery and drug targeting systems; and (vii) the possibility of controlling their release profile [137].

On the other hand, the applications of these peptides also have some limits, including: (i) their high conformational flexibility, causing a lack of receptor selectivity; (ii) the poor permeability of larger peptides through barriers; (iii) the possibility of reaching distant sites and provoking undesired side effects, reducing their efficacy at the implant site; (iv) their short half-life *in vivo*; and (v) their fast metabolism by the liver and kidneys [138]. However, several approaches can be used to overcome peptides’ limitations and improve their clinical applications [139] for the repair and/or regeneration of osteochondral defects. These approaches include structural modifications to improve their physicochemical properties, stability and half-life; conjugations with biocompatible polymers to increase their efficiency, reduce their liver and renal clearance and enhance their membrane permeability and target selectivity.

However, although further studies are needed before they can be used clinically for the repair and/or regeneration of osteochondral defects, the use of biomimetic peptides as alternative growth factors for OCTE appears promising.

## Figures and Tables

**Figure 1 ijms-23-07388-f001:**
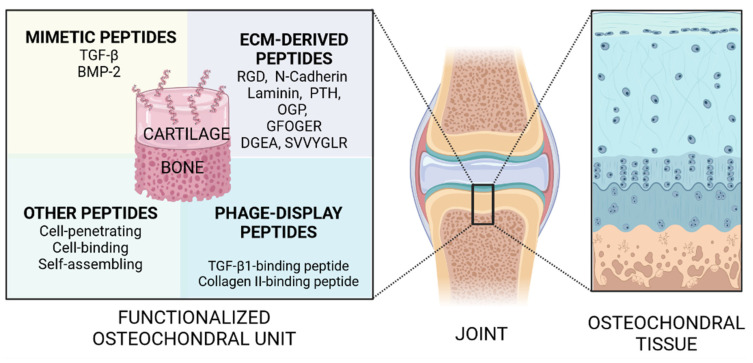
Schematic representation of peptides used in OCTE as a powerful tool to mimic the native osteochondral tissue.

**Figure 2 ijms-23-07388-f002:**
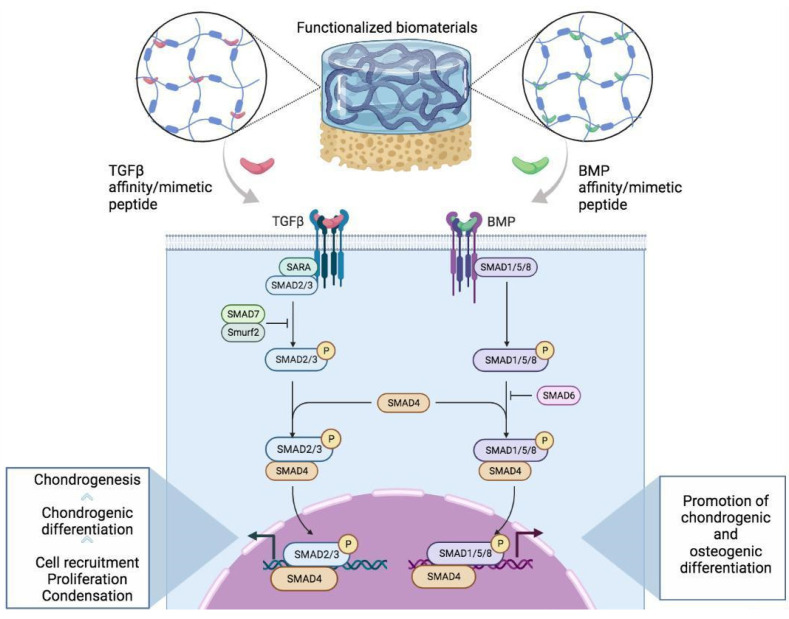
Representative scheme illustrating the molecular mechanisms of transforming growth factor beta (TGFβ) and bone morphogenetic protein (BMP) signaling-derived chondro- and osteoinductive peptides through the activation of SMAD-dependent signaling to induce/promote the chondrogenesis and/or osteogenesis of MSCs.

**Figure 3 ijms-23-07388-f003:**
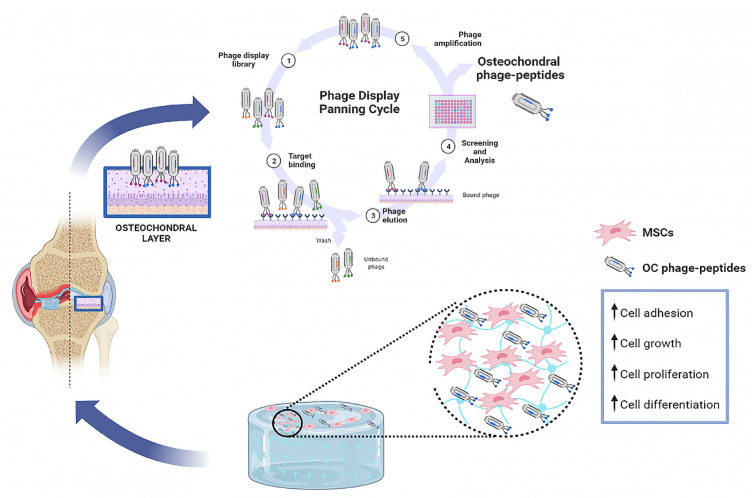
Functional peptide selection by means of the phage display (PD) approach to functionalize biomimetic scaffolds for OCTE. (1-5) Phage Display biopanning Cyclic representation. (1) Library construction; (2) Target capturing; (3) Elution of targets-specific phages; (4) screening and analysis; (5) Phage Amplification.

**Table 1 ijms-23-07388-t001:** List of the peptides involved in OCTE.

Name	Sequence	Function	Biomaterial	Peptide Amount	*In Vitro/In Vivo*	Ref.
Cytomodulins (CMs)	AA1-AA2-AA3…AAn (AA1 = A, N, L; AA2 = V, I; AA3 = A; AAn = Q, D, E, N)	Improve collagen I expression and promote wound healing *in vitro* and *in vivo*	-	CM-1, CM2, 0.1–1 μM	Human foreskin fibroblast (HFF) cells	[39]
			Poly(lactide-co-glycolide) (PLGA) microspheres	CM-1, 200 ng/mL	Human dermal fibroblast cells (HDFs)	[40]
			Co-assembly peptide amphiphile (PA)	5 or 10 mol% TGF-binding PA	Mesenchymal stem cells (MSCs) and rabbit model	[43]
			Chitosan (CS)	CS:peptide = 10:1 *w*/*w*, 10:2 *w*/*w*, 10:3 *w*/*w*	MSCs and mouse/rabbit model	[44]
			Gelatin methacryloyl (GelMA), GelMA/hydroxyapatite (HAp) (bilayered scaffold)	TGF-β1 peptide, via covalent linking, 50–400 μg/mL	MSCs and rat model	[45]
			GelMA	0.025 mM, TGF-β1-affinity peptide	Human umbilical cord mesenchymal stem cells (huMSCs) and rabbit model	[46]
			β-tricalcium phosphate (TCP)/PLGA–subchondral region. Poly(D,L-lactic acid-co-trimethylene carbonate)–cartilage region (bilayered scaffold)	20 μg of TGF-β1/collagen I (1 mL, 9 mg/mL), injected into cartilage region	Rat bone-marrow-derived mesenchymal stem cells (rBMSCs)	[47]
CK2.1	-	Drives chondrogenesis and induces the formation of cartilage	-	100 nM	C3H10T1/2 cells and murine model	[59]
			Hyaluronic acid hydrogel particles (HGPS)	HGPS (10 mg) in 10 mL PBS with CK2.1 (10 mg)		[60]
CK2.2 and CK2.3	-	Induce osteocalcin expression and mineral deposition	-	100 nM		[59]
OP-BMP-2	NSVNSKIPKACCVPTELSAI	Promote osteogenesis	Polyethylene terephthalate	BMPs, 10–3 M	MC3T3-E1 (pre-osteoblast-like) cells	[63]
pBMP-9	RKVGKASSVPTKLSPISILYK		Titanium	BMP-2, 2 mg/0.5 mL in PBS	MC3T3-E1 cells and beagle dogs	[64]
BMP-7	RTVPKPSSAPTQLNAISTLYF		-	BMPs, 0.02 to 200 mg/mL	MSCs	[65]
B2A	B2A2-K-NS	Increases cartilage proteoglycan synthesis and chondrogenesis *in vitro* and *in vivo*	-	0–1000 ng/mL	C2C12 cells and C3H10T1/2 cells	[66]
NLS-TAT	PKKKRKVKGRKKRRQRRRPPQ	Promotes chondrogenesis	-	4.10 μM	Rat precartilaginous stem cells (PSCs)	[75]
PepGen P-15	GTPGPQGIAGQRGVV	Promotes osteoblastic activity *in vitro* and bone regeneration *in vivo*	-	100 µg/mL	Human periodontal ligament fibroblast cells	[79]
			Phytogene HAp (Algipore^®^), TCP (Bio-Base^®^), bovine HAp (low temperature (T)) (Bio-Oss^®^), bovine HAp (high T) (Osteograf^®^), and bovine Hp (high T)	PepGen P-15^®^	Human osteoblast cells	[81]
			Grafting materials: BioOss, OsteoGraf N-300	PepGen P-15	Osteoblast cells	[82]
			Titanium	1 mM (dry dimethylformamide)	Preosteocyte MLO-A5 cells	[83]
			-	1000 μg/mL	Human osteoblast cells	[84]
			Anorganic bovine-derived mineral bound to a P-15 (ABM/P-15): ABM/P-15 carboxymethyl cellulose (CMC)-hydrogel graft and ABM/P-15 particulate graft	200 ng of P-15 per 1 g of ABM	Rabbit model	[85,86,87]
			Algae-derived hydroxyapatite/P-15 + 25% autologous bone	P-15 was adsorbed onto the HA surface	Pig model	[88]
			ABM (OsteoGrafs/N-700) and ABM/P-15 (PepGen	P-15^TM^, adsorbed		[89]
			ABM/P-15	-	Dog model	[90]
			ABM (OsteoGrafs/N-300) and ABM/P-15	P-15™, adsorbed	Human foreskin fibroblast cells	[91]
Peptide amphiphiles(PA)	-	Promote osteogenic differentiation *in vitro* and bone regeneration *in vivo*	Nanofibrous PA scaffold	PA-RGDS PA-DGEA	hMSCs	[95]
				Negatively charged PA with phosphoserine residues, negatively charged PA with serine residues, a positively charged PA with RGDS sequence, 10 mM		[96]
				OPD, 1%wt	hBMSCs	[97]
				10 mg/mL, PuraMatrix (RADA16-I)	Rabbit model	[99]
RGD	GRGDS, RGDS, YRGDS and c (RGDfk)	Improves cell adhesion and cartilage and bone tissue repair	Piranha activated borosilicate glass slides	200 μL of 20 μM solution of RGD-TAMRA, BMP-2- FITC or a mixture of both peptides	hBMSCs	[104]
			Maleimide functionalized polystyrene-block-polyethylene (PS-PEO) copolymer, spin-coated	20 μg/mL peptide (CGRGDS, CGGGRRETAWA, CGQAASIKVAVSADR or CGGEGYGEGYIGSR)	hMSCs	[105]
N-cadherin peptide	HAVDIGGGC	Promotes both early chondrogenesis of MSCs and late cartilaginous matrix production	Methacrylated hyaluronic acid	10 mol% of methacrylates	hMSCs and mouse model	[106]
			Self-assembly hydrogel	Self-assembly peptide (Ac-KLDLKLDLKLDL, KLD), N-cadherin mimetic self-assembly peptide (Ac-HAVDIGGKLDLKLDLKLDL, KLD-Cad), and scrambled control peptide (Ac-AGVIDHGKLDLKLDLKLDL, KLD-Scr), 0.5% (*w*/*v*)	MSCs	[107]
			Poly(glycolic acid)-poly(ethylene glycol)-poly(glycolic acid)-di(but-2-yne-1,4-dithiol) (PdBT) click conjugated with either chondrogenic “GGGHAVDI” or GGGGHKSP, mixed with poly(N-isopropylacrylamide-co-glycidyl methacrylate) (P(NIPAAm-co-GMA), (bilayered hydrogel)	-	MSCs and rabbit model	[108]
Laminin mimic peptides	IKVAV and YIGSR	Influence cartilage and bone regeneration	Maleimide functionalized PS-PEO copolymer, spin-coated	20 μg/mL (CGRGDS, CGGGRRETAWA, CGQAASIKVAVSADR or CGGEGYGEGYIGSR)	hMSCs	[110]
	Ln2-p3	Enhances the expression of osteogenic markers and increases ALP activity	Titanium	23 μg/cm^2^	Human osteosarcoma (HOS) cells	[111]
PTH 1–34	H-SVSEIQLMHNLGKHLNSMERVEWLRKKLQDVHNF-OH	Stimulates osteoblast proliferation, differentiation and prevents apoptosis	-	5 or 30 μg/kg of PTH (1–34)	Rat model	[117]
			-	PTH (1–34) in 0.9% saline at a daily dosage of 40 μg/kg body weight (BW), subcutaneously		[118]
			-	PTH (1–34) in saline solution 200 μL, subcutaneously	Mouse model	[119]
			-	10 μg/kg BW, daily, subcutaneously	Rabbit model	[120]
OGP and OGP10–14	YGFGG	Enhance osteoinductive potential	Bacte- rial cellulose-hydroxyapatite (BC-HA)	10^−9^ mol/L, adsorption	Mouse model	[121]
Collagen-mimetic peptide	GFOGER	Promotes bone regeneration and osseointegration	Titanium	20 μg/mL in Dulbecco’s PBS	Primary bone marrow stromal cells	[123]
	DGEA	Promotes osteogenic differentiation and bone formation	Collagen Type I	10 μg/mL	MC3T3-E1 cells	[124]
	SVVYGLR	Suppresses osteoclastogenesis and contributes to bone repair at the early stage	Atelocollagen sponge	10 μg	hMSCs and rat model	[126]

## Data Availability

Not applicable.

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
