# Peer review of "Multipotential Role of Growth Factor Mimetic Peptides for Osteochondral Tissue Engineering"

_ijms, 2022, doi:10.3390/ijms23137388_

Round 1
Reviewer 1 Report
In this review, the authors give an overview of recent literature underlining the specific peptides used for osteochondral repair strategies and summarize the current knowledge of the use of phage display peptides in osteochondral tissue regeneration.
Comments:
On page 8, reference 101, the authors should explain better how is designed the bi-layered modular hydrogel system.
The following recent reference should be added “Cryogenic 3D printing of heterogeneous scaffolds with gradient mechanical strengths and spatial delivery of osteogenic peptide/TGF-β1 for osteochondral tissue regeneration.Biofabrication. 2020 Mar 23;12(2):025030. doi: 10.1088/1758-5090/ab7ab5.”
Authors should also explain how biomaterials are biofunctionalized with peptides. Table 1 could be improved by giving more information about the functionalized biomaterial, the strategy used to entrap the peptide when a biomaterial is used, and the concentration of the peptides.
The conclusion part seems poor. This paper would be better if the authors could add a little bit more discussion about a future trend, and also explain the limitations of using peptides and what strategies or studies are needed in order to use them safely at the clinic for the repair and/or regeneration of osteochondral defects.
Author Response
Authors: Thank you very much for the potential publication of our manuscript. We sincerely thank the Reviewers for all the constructive criticisms and valuable comments, which were of great help in further revising and improving the manuscript. Accordingly, the manuscript has been revised and modified to take into account each suggestion. We submit the marked copies of the revised manuscript containing all tracked changes highlighted in yellow.
Reviewer: In this review, the authors give an overview of recent literature underlining the specific peptides used for osteochondral repair strategies and summarize the current knowledge of the use of phage display peptides in osteochondral tissue regeneration.
Reviewer: On page 8, reference 101, the authors should explain better how is designed the bi-layered modular hydrogel system.
Response: The authors thank the reviewer for the valuable comment. They added more information about the bi-layered modular hydrogel system. In particular, they added: “Articular cartilage repair was enabled by the bioconjugation of an N-cadherin peptide (NC) sequence while the subchondral bone repair was allowed by the conjugation of a glycine-histidine-lysine (GHK) sequence. In particular, the crosslinker poly(glycolic acid)-poly(ethylene glycol)-poly(glycolic acid)-di(but-2-yne-1,4-dithiol) (PdBT) was click conjugated with either chondrogenic NC (“GGGHAVDI”) or osteogenic GHK (“GGGGHKSP”) – peptide sequence, and then mixed with a suspension of thermoresponsive polymer, poly(N-isopropylacrylamide-co-glycidyl methacrylate) (P(NIPAAm-co-GMA), and MSCs to generate tissue-specific, cell-embedded hydrogel layers for cartilage and bone. The hydrogel layers were fabricated by mixing P(NIPAAm-co-GMA); either MSCs or no cells; and either PdBT (control), GHK/PdBT (osteogenic), or NC/PdBT (chondrogenic) in PBS on ice to obtain homogeneous pre-hydrogel suspensions, which were then poured into cylindrical Teflon molds (37 °C; 5% CO2, 1.5 h).”
Reviewer: The following recent reference should be added “Cryogenic 3D printing of heterogeneous scaffolds with gradient mechanical strengths and spatial delivery of osteogenic peptide/TGF-β1 for osteochondral tissue regeneration. Biofabrication. 2020 Mar 23;12(2):025030. doi: 10.1088/1758-5090/ab7ab5.”
Response: The authors thank the reviewer for the valuable suggestion. They added the suggested reference “Cryogenic 3D printing of heterogeneous scaffolds with gradient mechanical strengths and spatial delivery of osteogenic peptide/TGF-β1 for osteochondral tissue regeneration. Biofabrication. 2020 Mar 23;12(2):025030. doi: 10.1088/1758-5090/ab7ab5” in the paragraph 2.1. Transforming growth factor-β mimetic peptides. In particular, this part was added:
“Bilayered osteochondral scaffolds were also developed by Wang et al. through cryogenic 3D printing. In particular, the osteogenic peptide/β-tricalcium phosphate/PLGA water-in-oil composite emulsions were printed to obtain the porous subchondral layer, while poly(D,L-lactic acid-co-trimethylene carbonate) water-in-oil emulsions were processed into thermal-responsive cartilage frame on top of the subchondral layer. The osteogenic peptide (P24) had a sequence of KIPKA SSVPT ELSAI STLYL SGGC, with a purity of 98.12%. The cartilage frame was further functionalized with TGF-β1 loaded collagen I hydrogel to form the cartilage compartment. Results showed that the two layers were closely bonded together, showing excellent shear strength and peeling strength. Rat BMSCs exhibited high viability and proliferation in both layers. Moreover, gradient rBMSC osteogenic/chondrogenic differentiation was obtained in the osteochondral scaffolds.”
Reviewer: Authors should also explain how biomaterials are biofunctionalized with peptides. Table 1 could be improved by giving more information about the functionalized biomaterial, the strategy used to entrap the peptide when a biomaterial is used, and the concentration of the peptides.
Response: The authors thank the reviewer for the valuable suggestion. Accordingly, the authors added in Table 1, three columns in which information about the biomaterial, if used, the amount of the specific peptide, and the performed biological test (in vitro/in vivo) are reported. Furthermore, in the main text, this information was also added, in order to improve the clarity and the understanding.
Reviewer: The conclusion part seems poor. This paper would be better if the authors could add a little bit more discussion about a future trend, and also explain the limitations of using peptides and what strategies or studies are needed in order to use them safely at the clinic for the repair and/or regeneration of osteochondral defects.
Response: We thank the reviewer for the comment. Accordingly, we added a little discussion, in the section 4. Discussion and conclusions, to explain the limitations of using peptides and the approaches to overcome these limits in order to efficiently use them for the repair and/or regeneration of osteochondral defects.

Reviewer 2 Report
Authors have reviewed the works regarding the uses of peptides for osteochondral effects. The manuscript is well written. I suggest the following comments be addressed.
1. Several works have shown loading of the peptide in electrospinning fibers and use in bone regeneration. I felt missing that part and corresponding discussion.
2. The release profile of the peptides from biomaterial also is important to achieve target. Authors should discuss.
3. The limitations of the peptides in a given area should be discussed. Based on that what could be the solution to overcome limitations is suggested to add in revised work.
Author Response
Authors: Thank you very much for the potential publication of our manuscript. We sincerely thank the Reviewers for all the constructive criticisms and valuable comments, which were of great help in further revising and improving the manuscript. Accordingly, the manuscript has been revised and modified to take into account each suggestion. We submit the marked copies of the revised manuscript containing all tracked changes highlighted in yellow.
Reviewer: Authors have reviewed the works regarding the uses of peptides for osteochondral effects. The manuscript is well written. I suggest the following comments be addressed.
Reviewer: Several works have shown loading of the peptide in electrospinning fibers and use in bone regeneration. I felt missing that part and corresponding discussion.
Response: As kindly suggested by the reviewer, the authors added the outcomes of the most recent works regarding the electrospun fiber loading with bioactive peptides in bone regeneration approaches (paragraph 2.2 Bone morphogenetic protein - 2, pag. 7-8). Moreover, in the introduction section, the authors referred to the peptide immobilization on the electrospun fibers as a recent and common technique for a polymer surface biofunctionalization used in tissue engineering.
Reviewer: The release profile of the peptides from biomaterial also is important to achieve target. Authors should discuss.
Response: The authors thank the reviewer for the valuable suggestion. Indeed, this aspect is of paramount importance in order to obtain a successful tissue regeneration. Accordingly, the authors added the release profile behavior of specific peptides used to functionalize scaffolds and mentioned in the present review. In particular, they discussed the main results achieved in terms of peptide release kinetics, if the specific study was properly focused on this particular aspect. All the changes have been signed in yellow throughout the text.
Reviewer: The limitations of the peptides in a given area should be discussed. Based on that what could be the solution to overcome limitations is suggested to add in revised work.
Response: We thank the reviewer for the comment. Accordingly, we added a little discussion, in the section 4. Discussion and conclusions, to explain the limitations of using peptides and the approaches to overcome these limits in order to efficiently use them for the repair and/or regeneration of osteochondral defects.

Round 2
Reviewer 2 Report
The authors revised the manuscript satisfactorily. So it is ready to accept as it is in my opinion.